# Telomere Length in Pig Sperm Is Related to In Vitro Embryo Development Outcomes

**DOI:** 10.3390/ani12020204

**Published:** 2022-01-15

**Authors:** Jordi Ribas-Maynou, Yentel Mateo-Otero, Marina Sanchez-Quijada, Sandra Recuero, Ariadna Delgado-Bermúdez, Marc Llavanera, Marc Yeste

**Affiliations:** 1Biotechnology of Animal and Human Reproduction (TechnoSperm), Institute of Food and Agricultural Technology, University of Girona, 17003 Girona, Spain; yentel.mateo@udg.edu (Y.M.-O.); marinasanqui@gmail.com (M.S.-Q.); sandra.recuero@udg.edu (S.R.); ariadna.delgado@udg.edu (A.D.-B.); marc.llavanera@udg.edu (M.L.); Marc.yeste@udg.edu (M.Y.); 2Unit of Cell Biology, Department of Biology, Faculty of Sciences, University of Girona, 17003 Girona, Spain

**Keywords:** sperm, pig, telomere length, embryo development, in vitro fertilization

## Abstract

**Simple Summary:**

Understanding how gamete chromatin influences fertilization is highly important not only to improve animal production, but also to develop new biomarkers helping in the selection of those animals with higher fertility potential. In this regard, sperm telomere length has been pointed out as a putative biomarker in human infertility, but no studies have been conducted into its influence in pig fertility. Here, we determined that sperm telomere length is independent from the conventional sperm quality parameters and, through the production of in vitro embryos, we showed that it is indicative of the percentage of morulae and blastocysts, thus becoming useful to be used as biomarker in this species.

**Abstract:**

Telomere length has attracted much interest as a topic of study in human reproduction; furthermore, the link between sperm telomere length and fertility outcomes has been investigated in other species. This biomarker, however, has not been much explored in other animals, such as pigs, and whether it is related to sperm quality and fertility outcomes remains unknown. The present work aimed to determine the absolute value of telomere length in pig sperm, as well as its relationship to sperm quality parameters and embryo development. Telomere length was determined through quantitative fluorescence in situ hybridization (qFISH) in 23 pig sperm samples and data were correlated to quality parameters (motility, morphology, and viability) and in vitro fertilization outcomes. We found that the mean telomere length in pig sperm was 22.1 ± 3.6 kb, which is longer than that previously described in humans. Whilst telomere length was not observed to be correlated to sperm quality variables (*p* > 0.05), a significant correlation between telomere length and the percentage of morulae 6 days after in vitro fertilization was observed (r_s_ = 0.559; 95% C.I. = (−0.007 to 0.854); *p* = 0.047). Interestingly, this correlation was not found when percentages of early blastocysts/blastocysts (r_s_ = 0.410; 95% C.I. = (−0.200 to 0.791); *p* = 0.164) and of hatching/hatched blastocysts (r_s_ = 0.356; 95% C.I. = (− 0.260 to 0.766); *p* = 0.233) were considered. Through the separation of the samples into two groups by the median value, statistically significant differences between samples with shorter telomeres than the median and samples with longer telomeres than the median were found regarding development to morula (11.5 ± 3.6 vs. 21.8 ± 6.9, respectively) and to early blastocyst/blastocysts (7.6 ± 1.4 vs. 17.9 ± 12.2, respectively) (*p* < 0.05). In the light of these results, sperm telomere length may be a useful biomarker for embryo development in pigs, as sperm with longer telomeres lead to higher rates of morulae and blastocysts.

## 1. Introduction

Telomeres, ribonucleoprotein structures located at the end of chromosomes, are involved in the maintenance of genomic integrity and stability and are crucial for gamete generation [1]. Telomere sequences, which are similar in all vertebrates, are composed of repetitive, non-coding, double-stranded sequences of thousands of hexanucleotide DNA repeats (TTAGGG)_n_ coupled to specialized proteins of the shelterin complex and non-coding RNAs. Telomeres prevent the chromosome ends to be erroneously recognized as double-strand breaks [2].

Despite the conserved sequence of telomeres among mammals, the number of repetitions may be highly variable, the different tissues and cells from the same individual having distinct telomere lengths [3]. Moreover, factors such as age, sex or exposure to oxidative stress and genotoxic agents including radiation and chemical agents have been purported to influence that length, leading to telomere attrition [4]. In effect, previous investigations showed that an increase in reactive oxygen species (ROS) can bring about misbalanced ratios of ROS/antioxidants and drive to DNA damage through the alteration of DNA bases, like oxidation of guanine to 8-oxoguanine (8-oxoG), which ends up as an excised base [5]. As telomeres consist of G-rich sequences, they could show higher susceptibility to oxidative radicals and be more sensitive to the accumulation of 8-oxoG, thus underlying the disruption of telomeric proteins and the inhibition of telomerase, which results, in turn, in the shortening, dysfunction, and instability of telomeres [6]. 

While telomere length in humans has been described to be typically between 5 and 20 kb [4,7,8,9], previous studies demonstrated that male germ cells usually have longer telomeres than their somatic counterparts, because higher expression of telomerase has been observed in undifferentiated spermatogonia [10]. When referring to the later stages of spermatogenesis, however, it is well known that telomerase activity decreases progressively until reaching complete inactivation in mature sperm [11]. Sperm cells from a single individual, therefore, may depict different telomere lengths on account of their exposure to ROS and also in relation to the specific telomerase activity during early spermatogenesis stages [12]. Earlier research in humans showed that telomere length of ejaculated sperm may be associated to male reproductive disorders [4,12], and that telomere shortening could be related to the production of sperm with morphological alterations, chromosomal rearrangements, immaturity features, and high DNA damage [13,14,15]. Regarding its relation to clinical outcomes, inconsistent data in the literature are found. On the one hand, two studies conducted in infertile men showed that patients with short telomeres are incapable or less prone to produce good quality embryos, compared to those with longer telomere lengths [13,16]. On the other hand, nevertheless, other authors found no statistically significant differences in reproductive outcomes between sperm with long and those with short telomere length from donor samples [17]. 

In production animals, such as pigs, very little natural mating occurs in farms and most of the litters are born after artificial insemination using liquid-stored or cryopreserved sperm [18]. In this regard, several years of artificial genetic selection of boars producing the highest sperm quality has led to highly improved fertilization rates [19]. To date, only a few studies have been focused on analyzing pig telomere length, showing that it ranges between 10 and 30 kb in somatic tissues, which is longer than the figures obtained in humans [20,21]. With respect to mature sperm, Telomere Restriction Fragment (TRF) analysis has revealed that telomere length ranges between 13 and 44 kb [20,21]; these values, however, need to be confirmed through other methods. Moreover, and to the best of our knowledge, no research has been hitherto conducted to evaluate whether sperm telomere length is related to in vitro fertilization outcomes and embryo development. 

In the light of all the aforementioned, the aims of the present study were: (a) to characterize telomere length in pig sperm through the adaptation of an improved method based on quantitative fluorescence in situ hybridization (qFISH), and (b) to evaluate the relationship of that sperm telomere length with semen quality, in vitro fertilization (IVF) outcomes and embryo development.

## 2. Materials and Methods

### 2.1. Reagents

Unless stated otherwise, all reagents were purchased from Sigma-Aldrich (St. Louis, MO, USA).

### 2.2. Semen Samples and Ethics

A sperm sample from each of the sexually mature Piétrain boars included in the study (*n* = 23) was collected. Animals were housed in a local farm operating under standard conditions to produce commercial, artificial insemination doses (Servicios Genéticos Porcinos, S.L.; Roda de Ter, Barcelona, Spain). Samples were collected from February to March 2021, and the age of boars ranged between 18 and 30 months. All samples were used to evaluate the relationship between telomere length and sperm quality. Thirteen randomly selected samples were utilized to assess the link between telomere length and IVF outcomes. The gloved-hand method was used to collect the animals, and ejaculates were immediately diluted to a concentration of 3.3 × 10^7^ sperm/mL in a long-term extender (Vitasem, Magapor S.L; Zaragoza, Spain). Thereafter, samples were brought to the laboratory at 17 °C and, upon arrival, semen quality was evaluated. Samples were kept at 17 °C until the next day, when IVF experiments were conducted.

Animals were not directly manipulated for the study, since samples were provided commercially by the farm, operating under the ISO certification (ISO-9001:2008) and following the current regulations on Animal Welfare, Health, and Biosafety issued by the Department of Agriculture, Livestock, Food, and Fisheries (Generalitat de Catalunya, Spain). Since no animal was manipulated for the study because samples were purchased from the farm, there was no need for specific ethical approval to conduct the experiments.

### 2.3. Telomere Length Analysis through Quantitative Fluorescent In Situ Hybridization (qFISH) 

The method to evaluate telomere length was the one previously published in human sperm [22,23] and adapted to pigs with slight modifications from our lab. This technique is based on qFISH using sperm previously submitted to chromatin decondensation (haloes), which provides higher sensitivity in the analysis of sperm telomere length. 

#### 2.3.1. Sperm Chromatin Dispersion

Initially, samples were centrifuged at 300× *g* for 5 min and then washed in PBS. Samples were adjusted to a concentration of 10^6^ sperm/mL and mixed 1:2 (*v*/*v*) with 1% liquefied low melting point agarose. Thereafter, 6.5 µL of the mix was placed onto an agarose pre-treated slide, covered with an 8-mm round coverslip and cooled at 4 °C for 5 min. After carefully removing the coverslips, slides were submerged in lysis solution 1 (0.8 M DTT, 0.8 M Tris, 1% SDS, and pH = 7.5) for 30 min, in lysis solution 2 (0.4 M Tris, 0.4 M DTT, 2 M NaCl, 50 mM EDTA, 1% Tween20, and pH = 7.5) for 30 min, and in lysis solution 3 (0.4 M Tris, 0.4 M DTT, 2 M NaCl, 50 mM EDTA, 1% Tween20, 100 µg/mL proteinase K, and pH = 7.5) for 3 h. These lysis protocols have recently been described to be required for the complete decondensation of pig sperm chromatin and proper formation of haloes [24]. Finally, slides were washed in PBS for 5 min and dehydrated in an ethanol series (70%, 90%, and 100%) for 2 min each.

#### 2.3.2. Quantitative Fluorescent In Situ Hybridization (qFISH)

Slides in which sperm chromatin decondensation was performed were fixed with 10% formaldehyde for 12 min, washed with PBS and dehydrated in an ethanol series (70%, 90%, and 100%) for 2 min each. Afterwards, DNA was denatured through incubation in 0.5 M NaOH for 4 min, again dehydrated in ethanol series and dried horizontally. Next, 15 µL of the Peptide Nucleic Acid (PNA) probe (TTAGGGTTAGGGTT, TelG, Panagene; Korea) at a final concentration of 0.8 µg/mL in hybridization buffer (10 mM Na_2_HPO_4_, 10 mM NaCl, 20 mM Tris, 70% formamide, and pH = 7.4) was denatured at 72 °C for 6 min and subsequently poured on the top of each sample. Following this, slides were covered with parafilm and placed for hybridization in a small, damp container at room temperature for 1 h in the dark. 

After hybridization, parafilm was carefully removed and three washings were conducted: first, in PBS + 0.1% Tween20 at room temperature and darkness for 2 min; second, in the same solution at 48 °C and darkness for 20 min; and, finally, with 2× SSC + 0.1% Tween 20 at room temperature and darkness for 2 min. Slides were subsequently dehydrated in an ethanol series (70%, 90%, and 100%), dried horizontally, and counterstained with DAPI Antifade Mountant (Thermo Fisher Scientific; Waltham, MA, USA). Slides were stored at 4 °C until microscope images were captured, which was performed on the following day.

#### 2.3.3. Image Captures and Analysis

An epifluorescence microscope (Axio Imager.Z1, Carl Zeiss AG; Oberkochen, Germany) equipped with a camera (AxioCam, Carl Zeiss AG; Oberkochen, Germany) was used to visualize and capture fluorescence intensities. Images of telomeres (FITC) from 20 individual haloes were acquired with the AxioVision software (AxioVs40 V 4.6.1.0, Carl Zeiss AG; Oberkochen, Germany) at 1000× magnification and after 1 s of exposure, as determined by the setting up described below. This time of exposure enabled us to reduce the background noise compared to shorter times. Images were saved in.TIFF format (16 bits/channel). Fluorescence intensity from the FITC channel (Figure 1B) was analyzed using the TFL-Telo v2 software (British Columbia Cancer Centre; Vancouver, Canada), the threshold value to identify only the specific hybridization signals being set up. The software retrieved the fluorescence intensity for each telomere and the area of the captured telomeric signal. Intensity divided by area was used as a standard parameter for the measurement of telomere length. Previous works have used similar methods to obtain a relative value of telomere length [22,23].

#### 2.3.4. Setting Up the Exposure Time for the Evaluation of Fluorescence Intensity 

In order to set up the capture time conditions to avoid the saturation of the hybridization signal, a qFISH was conducted in two sperm samples following the previously described protocol. Images of at least 10 individual sperm were taken under 1000× magnification, at increasing exposure times (from 0.25 s to 1.75 s). The mean saturation curve was compiled to determine an exposure time where the log of the fluorescence intensity depicted linearity, thus preventing obtaining a saturated or dim signal. 

#### 2.3.5. Absolute Telomere Length Estimation 

To estimate the absolute telomere length in kb, a previously published protocol, where a relationship between fluorescence intensity and length was validated using 1301 cells, was set up and adapted to our conditions/samples [22,23]. Briefly, an analysis of fluorescence intensity and signal area of SPHERO Calibration Particles (Rainbow Calibration, 8 peaks, 3.0–3.4 µm (FITC); Spherotech, CA, USA) was performed using the same capturing conditions and the TFL-Telo software parameters defined above, and only the spheres falling in the range of the telomere intensity were used for calibration (Appendix A). The SPHERO calibration particles are routinely used for calibration of flow cytometers, and contain a known number of attached fluorochrome molecules. A linear equation correlating fluorescence intensity divided by the signal area and number of fluorochrome molecules was thus worked out. Knowing that the PNA probe used for hybridization was 14 bp in length (TTAGGGTTAGGGTT) and was attached to one molecule of fluorochrome, we were able to extrapolate fluorescence intensity into absolute telomere length using the formula described in the results section. 

### 2.4. Sperm Quality Parameters

#### 2.4.1. Sperm Motility 

Sperm motility parameters were analyzed using a commercial computer-assisted system (ISAS software, Integrates Sperm Analysis System V1.0; Proiser S.L.; Valencia, Spain) connected to an Olympus BX41 microscope (Olympus; Tokyo, Japan) with a negative phase contrast field (Olympus 10× 0.30 PLAN objective, Olympus). In order to assess sperm motility, 3 µL of each sample pre-warmed to 38 °C was loaded into a Leja20 counting chamber (Leja Products BV; Nieuw-Vennep, The Netherlands). Motility was assessed capturing different fields at 25 frames/s and 100× magnification until at least 500 sperm per replicate were counted. Two technical replicates were examined. 

Parameters recorded through the automated software were: percentages of progressive motility (STR ≥ 45%), total motility (VAP ≥ 10 µm/s), curvilinear velocity of sperm (VCL, sperm progression along the trajectory; µm/s); sperm straight-line velocity (VSL, the straight trajectory per second; µm/s); average path velocity (VAP, the mean trajectory per second; µm/s); linearity coefficient (LIN, (VSL/VCL) × 100; %); straightness coefficient (STR, (VSL/VAP) × 100; %); wobble coefficient (WOB, (VAP/VCL) × 100; %); the amplitude of lateral head displacement (ALH, mean amplitude of the head lateral oscillatory movement; µm) and frequency of head displacement (BCF; the sperm head lateral oscillatory movements per second; Hz).

#### 2.4.2. Sperm Morphology

For sperm morphology, incubation in 2% formaldehyde at room temperature was conducted for 5 min and sperm were classified as morphologically normal or abnormal (including proximal or distal droplets, head abnormalities and tail abnormalities). The SCA Production software Sperm Class Analyzer Production, 2010; Microptic S.L., Barcelona, Spain) was used to aid with this classification. Two replicates assessing 100 sperm each were analyzed per sample. 

#### 2.4.3. Sperm Viability

The assessment of plasma membrane integrity through the Live/Dead Sperm viability kit (Molecular Probes, Eugene, OR, USA) was used to determine sperm viability, following the protocol of Garner and Johnson [25] with minor modifications. In brief, sperm were diluted to a final concentration of 1 × 10^6^ in pre-warmed PBS and stained with SYBR-14 (final concentration: 32 nM) at 38 °C in the dark for 10 min. Following this, Propidium Iodide (PI; final concentration; 7.5 µM) was added and samples were incubated at the same conditions for further 5 min. Subsequently, samples were analyzed with a CytoFLEX cytometer (Beckman Coulter; Fullerton, CA, USA) recording 10,000 events. SYBR-14 fluorescence was detected with the FITC channel (525/40) and PI fluorescence was collected through the PC5.5 channel (690/50). Both fluorochromes were excited with a 488-nm laser, and no compensation was needed. The percentage of events falling into the viable gate (SYBR-14^+^/PI^−^) with respect to the total sperm was recorded and used for statistical analyses.

#### 2.4.4. Sperm DNA Damage 

Sperm DNA fragmentation was assessed through the alkaline Comet assay, following the protocol described for pig sperm [26]. Initially, samples were prepared following the protocol described in Section 2.3.1 for the evaluation of chromatin dispersion. After submerging slides into the third lysis solution, they were incubated in an alkaline solution (0.03 M NaOH, 1 M NaCl) at 4 °C for 5 min. Following this, samples were electrophoresed at 1 V/cm for 4 min in an alkaline buffer (0.03 M NaOH; pH = 13). Finally, slides were incubated in a 0.4 M Tris-HCl solution (pH = 7.5) for 5 min, and dehydrated in an ethanol series (70%, 90%, and 100%). Comet staining was conducted following the addition of 5 µL of 1× Safeview DNA stain (NBS biologicals, Huntington, UK); slides were covered with a 20 × 20 coverslip. Images of at least 100 comets per sample were captured at 100× magnification under a Zeiss Imager Z1 epifluorescence microscope (Carl Zeiss AG, Oberkochen, Germany), at a resolution of 1388 × 1040 pixels. The exposure time was adapted for each image to avoid overexposure of Comet heads. All images were automatically analyzed using the CometScore v2.0 software, after adjusting the background intensity to correctly visualize Comet heads and tails. After removing the captures containing debris particles interfering to the analysis and revising the misanalysed Comets, a mean olive tail moment (OTM) was obtained for each sample as an indicative parameter of the amount of DNA damage. 

### 2.5. In Vitro Fertilization (IVF)

In vitro fertilization experiments were conducted using a subgroup of 13 samples randomly selected from the aforementioned boars. As defined below and in Appendix A, these samples were split into two categories: samples with telomeres shorter than the median group (samples 1, 2, 3, 4, 5, and 7; *n* = 6), and those with telomeres longer than the median group (samples 12, 14, 15, 19, 20, 22, and 23; *n* = 7). Each ejaculate from a different male was treated as a biological replicate, and used for the IVF experiment. For each of these 13 replicates, 40 MII oocytes were fertilized, and embryo development was assessed. We used this number of oocytes because 35 oocytes was statistically determined as the minimum number to determine an average blastocyst rate of 13% ± 1.3% [27], and 40 was larger than this figure. Previous studies in pigs also used similar pools of mature oocytes to conduct IVF experiments [28,29].

Ovaries were obtained from pre-pubertal sows at a local slaughterhouse dedicated to meat industry (Frigorifics Costa Brava; Riudellots de la Selva, Girona, Spain). Immediately after collection, ovaries were preserved in 0.9% NaCl supplemented with 70 µg/mL kanamycin at 35 °C and transported to the laboratory. Upon arrival, >250 cumulus-oocyte complexes (COC) were collected from follicles through aspiration. Thereafter, groups of 50 COCs were relocated into four-well multi-dishes (Nunc, TermoFisher; Waltham, MS, USA) and incubated in 500 µL of pre-equilibrated maturation medium (TCM-199 supplemented with 0.1% (*w*/*v*) PVA, 10 ng/mL EGF, 0.57 mM cysteine, 75 µg/mL penicillin-G potassium, and 50 µg/mL streptomycin sulphate), with an additional supplementation of 10 IU/mL of human Chorionic Gonadotropin (hCG, Veterin Corion, Divasa Farmavic S.A.; Gurb, Barcelona, Spain) and 10 IU/mL of equine Chorionic Gonadotropin (eCG, Folligon, Intervet International B.V.; Boxmeer, The Netherlands), for 22 h. At this point, the medium was replaced by 500 µL of pre-equilibrated maturation medium without hormone supplementation, and oocytes were incubated for further 22 h. Oocytes that successfully achieved in vitro maturation and presented the best morphology were deemed eligible to be fertilized. 

Before the fertilization step, matured oocytes were pooled and denuded in Dulbecco’s PBS (Gibco, TermoFisher, Waltham, MS, USA). Next, oocytes were grouped randomly and transferred into 50-µL drops of pre-equilibrated in vitro fertilization medium (modified Tris-buffered medium [30]) supplemented with 1 mM caffeine. Simultaneously, semen samples were diluted to a final concentration of 1000 sperm cells per oocyte with in vitro fertilization medium. 

For fertilization (day 0), oocytes (at least 40 per sample) and sperm (1000 sperm per oocyte) were co-incubated for 5 h. Presumptive embryos were washed and transferred into four-well multi-dishes (40 embryos/well) containing 500 µL of NCSU23 supplemented with 0.3 mM pyruvate and 4.5 mM lactate. After 48 h, cleaved embryos were counted to calculate the Fertilization Rate (day 2). Embryos were moved to fresh NCSU23 medium supplemented with 0.4% BSA and 5.5 mM glucose, where they were cultured up to day 6. At that time, embryos were classified following the criteria reported by Balaban and Gardner [31] in non-developing embryos, morulae, early blastocysts/blastocysts and hatching/hatched blastocysts. Embryo development (day 6) was expressed as percentages of each of these categories, considering the total number of oocytes included in the experiment.

Oocyte maturation, in vitro fertilization and embryo culture were carried out at 38.5 °C under a continuously humidified atmosphere and 5% CO_2_ in air.

### 2.6. Statistical Analysis 

Statistics Package for Social Sciences (SPSS, Ver. 25, IBM Corp; Armonk, NY, USA) was used to conduct statistical analyses. GraphPad Prism Software (GraphPad; San Diego, CA, USA) was used to create graphs. Normal distribution was assessed using the Shapiro-Wilk test, and the Levene test was used to verify the homogeneity of variances. Correlation with seminal parameters was calculated using the Pearson test, whereas correlations with IVF outcomes were determined using the non-parametric Spearman test. 

In order to compare sperm with longer and shorter telomeres and assess the effect of telomere length, samples were divided taking the median value of telomere length as a cut-off value. Therefore, samples with a mean sperm telomere length higher than the median were considered longer (Longer telomere length), whereas samples with a mean sperm telomere length shorter than the median were considered shorter (Shorter telomere length). Samples with telomeres longer than the median ranged between 10.38 kb and 21.94 kb, whereas samples with telomeres shorter than the median ranged between 22.04 kb and 29.17 kb. For sperm morphology, viability and DNA damage, the Mann–Whitney U test was used to compare samples with longer or shorter telomere length. For sperm motility, a two-way ANOVA with Bonferroni correction for multiple comparisons was run. For IVF results, the Mann–Whitney U test was used to compare Fertilization rate at day 2, and a two-way ANOVA with Bonferroni correction for multiple comparisons was conducted for day 6 parameters. For all statistical tests conducted, a significance level of 95% was established (*p* ≤ 0.05).

## 3. Results

### 3.1. Setting Up the Evalution of Sperm Telomere Length

As described in the Methods (Section 2.3.4), different exposure times were tested to set up the imaging conditions (from 0.25 s to 1.75 s). Through the representation of the log (Fluorescence Intensity/Signal Area), the exposure time was determined to be between 0.5 s and 1.5 s. Following this, microscope captures were established at 1 s in all cases. Thereafter, and in order to determine the relationship between fluorescence intensity and fluorochrome molecules, the analysis of fluorescent spheres led to a linear correlation between these two variables (*r* = 0.997; *p* = 0.045; *n* = 36) which, in turn, could be extrapolated to absolute telomere length in our hybridization system. Thus, the final equation relating fluorescence intensity (in arbitrary units) to absolute telomere length (in kb) was: Telomere length (kb) = 1.625 × Fluorescence intensity (I/A) − 2.812

### 3.2. Relationship between Sperm Telomere Length and Quality Parameters

Figure 1 shows an example of a sperm halo that was used to conduct qFISH. The analysis of 23 pig sperm samples showed that telomere length ranged from 10 kb to 29 kb, with a mean and standard deviation of 22.14 ± 3.63 and a median of 21.99 kb. The absolute sperm telomere length of all samples is shown in Figure 2 and Appendix A. 

Sperm quality parameters of different samples and values for shorter and longer telomere length are shown in Table 1. When the median of telomere length values was used to separate samples into two groups (longer or shorter than the median), no differences in sperm motility, viability, or morphology between these two groups were found (*p* > 0.05) (Table 1). 

Correlation analysis between sperm telomere length and quality parameters showed no statistically significant correlation between telomere length and sperm motility, as depicted in Table 2. No correlation between sperm telomere length and viability, morphology, or DNA damage was observed (*p* > 0.05).

### 3.3. Relationship between Sperm Telomere Length and In Vitro Fertilization Outcomes

In vitro fertilization outcomes are shown in Table 3 and Figure 3. A total of 525 mature oocytes were used for fertilization (240 for the group with shorter telomeres, and 285 for the group with longer telomeres). A total of 84 embryos reached morula (27 and 57 for the group with shorter and longer telomere length, respectively), and 67 embryos reached early blastocyst/blastocyst stage (18 and 49 for the group with shorter and longer telomere length, respectively). Comparison between samples with shorter and longer telomere length showed that percentages of morulae and early blastocysts/blastocysts at day 6 post-fertilization were significantly higher in the latter than in the former *(p =* 0.018, and *p* = 0.018, respectively). 

Correlations between sperm telomere length and IVF outcomes are shown in Table 4. Statistically significant correlations between sperm telomere length and the percentage of morulae at day 6 post-fertilization (*r* = 0.559; *p* = 0.047) were found, but not between telomere length and the percentages of early blastocysts/blastocysts or hatching/hatched blastocysts (*p* > 0.05). 

## 4. Discussion

In the present study, we successfully estimated, for the first time, the length of individual telomeres in pig sperm using a qFISH methodology applied to decondensed haloes. As previously reported, the application of a decondensation step in pig sperm is required to achieve complete chromatin unpackaging and allows obtaining proper haloes [24]. This was combined with a method to estimate telomere length that was previously established in humans [22] and was adapted to pig sperm herein. This method uses flow cytometry calibration particles and enables the extrapolation of fluorescence intensity into absolute DNA length. Figure 1 depicts all these features, and demonstrates that: a) we were able to decondense pig sperm chromatin completely (Figure 1A) [24] and b) telomere signals were specific (Figure 1B), in a similar fashion to that reported by previous works in humans [23]. Through this approach, we determined that telomere length could be an independent quality biomarker in pig sperm because, despite not being correlated with quality parameters (motility, morphology, viability, or DNA damage), IVF outcomes (% morulae and %blastocysts) were higher in sperm samples with longer than in those with shorter telomere length (taking the median value as a split point between these two groups). 

### 4.1. Telomere Length in Pig Sperm

We evaluated, for the first time, telomere length in pig sperm using a qFISH approach. Results showed that, in pigs, sperm telomere length ranges from 10 to 29 kb, with a mean of 22.14 kb. Despite this broad range, it is worth mentioning that 20/23 samples (87%) ranged between 18 and 25 kb, just one sample being below 18 kb and two above 25 kb. To the best of our knowledge, only two previous studies estimated telomere length in pig sperm. These works used the telomere restriction fragment method (TRF) and found the range to be between 13 and 44 kb [20,21]. Using another technique, therefore, we could narrow the range determined before. This supports that further research should address whether differences in the detection of telomere length exist between methods (TRF, qFISH and qPCR), and also between different breeds and ages of the animals. 

Compared to other species, sperm telomere length in pigs appears to be longer than in humans (10 to 20 kb) [8,32,33] and cattle (15 kb) [21], but similar to dogs (11 to 27 kb) [34]. In the light of these findings, and as mentioned before, one should take into consideration that differences between methods (i.e., qFISH and TRF) leading to different outcomes may exist. The sperm telomere length found herein is shorter than that reported by Fradiani et al. [20], who performed the TRF method. The fact that the telomere length range is narrower in qFISH than in TRF is not an exception of pig species, as other studies reported the same trend in normospermic men [35,36]. While these differences could be attributed to a limitation of TRF, which may include subtelomeric regions and thus overestimate the number of kilobases, this does not occur when qFISH is used.

### 4.2. Relationship between Sperm Telomere Length and Quality Parameters

For the second aim of the present work, we studied the relationship between pig sperm telomere length and quality measured as motility, morphology, viability, and DNA damage. To the best of our knowledge, sperm telomere length has been studied as a possible biomarker for human [12,13] and bull (in)fertility [37], but no reports have been published regarding pig sperm telomere length and sperm quality or fertility parameters. The results obtained in the present research showed no statistically significant correlation of sperm telomere length with motility, viability, morphology, or DNA fragmentation. In addition, the comparison between samples with shorter and longer telomere length did not show significant differences in these sperm quality parameters. Telomere length has been suggested to be an indicator of normal spermatogenesis, given the crucial role of telomeres during meiosis and their contribution to genome integrity [38]. In this regard, despite the lack of previous results in pigs, investigations conducted in humans and a work conducted in bull sperm reported a relationship between sperm telomere length and quality parameters such as motility, morphology, sperm count, and viability [13,15,37]. This topic, however, is the subject of controversial results, as other studies found no correlation between sperm telomere length and motility parameters in human sperm samples [9,36]. In the light of these previous reports, one should take into account that differences between men cohorts may exist, because they may include both fertile and infertile individuals. Pig sperm samples, nevertheless, usually depict less variability than their human counterparts, possibly due to the strong genetic selection that the former species has suffered in terms of fertility [39]. In effect, among these farm animals, one hardly ever observes infertile individuals; thus, we tested correlations using boars with normal sperm quality which is usually related to normal artificial insemination outcomes, this being a substantial difference compared to human samples. This could also apply to telomere length; as samples used in this research were commercial and came from AI-boars, they fell in the range of standard telomere length. Yet, a recent study in bovine showed that sperm telomere length measured through qPCR is associated to sperm quality parameters, the bulls showing shorter telomere length being unsuitable as sires [37]. Hence, differences may not only be attributed to the method or cohort, but also to species, as a distinct selection of animals may have been performed. Additionally, it has been suggested that telomere length may be affected by other parameters such as age or oxidative stress [9,32]. Although this could not be assessed in the present study, further research in this realm is much warranted. 

### 4.3. Relationship between Sperm Telomere Length and IVF Outcomes 

For the third aim of the present study, we tested whether sperm telomere length has any impact on in vitro fertility outcomes. We obtained a significant positive correlation between sperm telomere length and the percentage of morulae at six days post-fertilization, showing that oocytes fertilized using sperm with longer telomeres resulted in a larger number of good quality embryos. It is worth noting that, while early blastocysts/blastocysts depicted differences between samples with longer and shorter telomere length (Figure 3), the correlation observed in morulae was lost when more advanced stages of embryo development were evaluated (i.e., blastocysts; Table 4). Previous works showed that telomerase activity may be increased after the morula stage, thus providing embryos with higher capacity to reach the blastocyst stage [40]. In humans, earlier research relating IVF outcomes to sperm telomere length reported that samples with an abnormal sperm telomere length resulted in a pregnancy rate of 0% [13]. Although most of the animals included in our study depicted normal sperm telomere length, differences in fertility outcomes between boars with longer sperm telomeres and those with shorter sperm telomeres could be aligned to what observed in humans. Further studies focused on analyzing the relationship between sperm telomere length and telomerase activity in embryos would help better understand the telomere dynamics along preimplantational development, especially in morula and blastocyst stages.

## 5. Conclusions

In conclusion, we were able to successfully determine sperm telomere length in pigs and to study the relationship between sperm quality variables and embryo development using qFISH in decondensed haloes. Although we found that telomere length is not an indicator of sperm quality, we advise that it could be a suitable biomarker of preimplantational embryo development after IVF in pigs, as sperm with longer telomeres led to higher rates of morulae and early blastocysts + blastocysts at Day 6.

## Figures and Tables

**Figure 1 animals-12-00204-f001:**
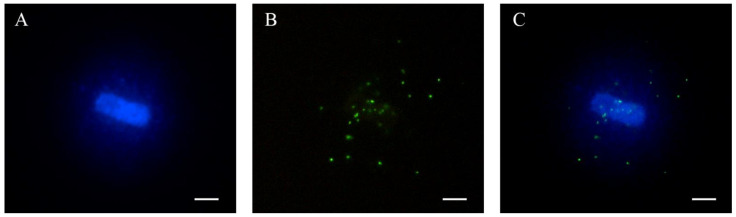
Representative images of a pig sperm halo (**A**) stained with DAPI, (**B**) telomeres labeled through the qFISH method and (**C**) merged images. Scale bar represents 5 µm.

**Figure 2 animals-12-00204-f002:**
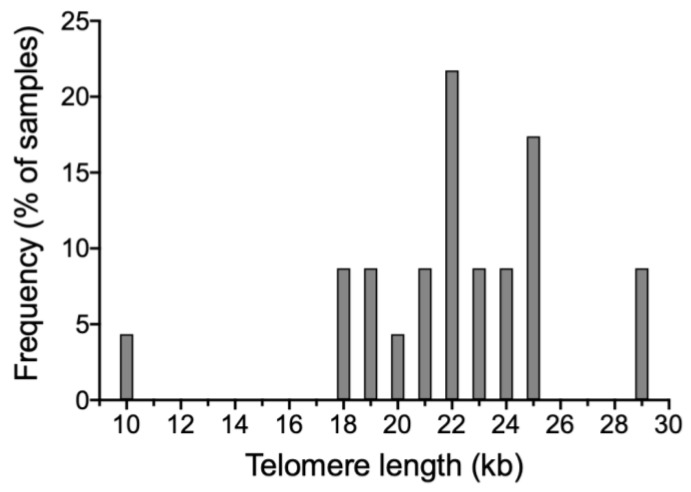
Distribution of frequencies for sperm telomere length (kb) in different individuals.

**Figure 3 animals-12-00204-f003:**
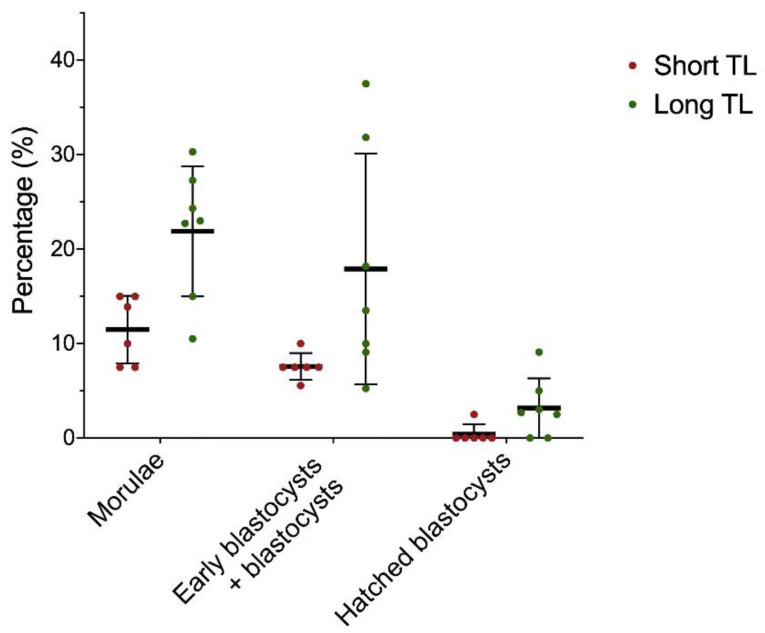
Percentages of morulae, early blastocysts/blastocysts, and hatching/hatched blastocysts at Day 6 post-fertilization for sperm samples with shorter and longer telomere length.

**Table 1 animals-12-00204-t001:** Telomere length and sperm quality parameters for all analyzed samples, and for samples classified into two groups based on their telomere length (shorter or longer than the median). *p*-values indicate statistical differences between groups with shorter and longer telomere length.

Parameter	Mean ± SD			*p*-Value
	All Samples	Group with Shorter Telomere Length	Group with Longer Telomere Length	
	*n* = 23	*n* = 11	*n* = 12	
Telomere length (kb)	22.1 ± 3.6	19.4 ± 3.3	24.7 ± 2.3	N/A
Progressive motility (%)	73.2 ± 7.9	71.0 ± 8.5	75.1 ± 7.1	>0.999
Non-progressive motility (%)	18.9 ± 8.6	19.4 ± 6.2	18.4 ± 10.5	>0.999
Fast sperm (%)	67.2 ± 16.6	62.3 ± 16.1	71.7 ± 16.3	0.277
Intermediate sperm (%)	17.8 ± 12.5	20.2 ± 11.3	15.7 ± 13.6	>0.999
Slow sperm (%)	7.0 ± 4.7	8.0 ± 4.8	6.0 ± 4.5	>0.999
Static sperm (%)	8.0 ± 7.9	9.6 ± 9.4	6.6 ± 6.2	>0.999
VCL (µm/s)	64 ± 13.6	61 ± 13.2	66.7 ± 13.9	>0.999
VSL (µm/s)	37.5 ± 9.6	35.2 ± 6.5	39.7 ± 11.7	>0.999
VAP (µm/s)	48.1 ± 9.5	45.7 ± 8.9	50.3 ± 9.8	>0.999
LIN (VSL/VCL) (%)	59.2 ± 11.2	58.7 ± 8.1	59.9 ± 13.9	>0.999
STR (VSL/VAP) (%)	77.8 ± 10.9	77.6 ± 9	78.1 ± 12.9	>0.999
WOB (VAP/VCL) (%)	75.7 ± 6.2	75.4 ± 5.4	76 ± 7	>0.999
ALH (µm)	2.3 ± 0.5	2.3 ± 0.4	2.5 ± 0.6	>0.999
BCF (Hz)	7.8 ± 1.3	7.8 ± 1.2	7.8 ± 1.4	>0.999
Viability (%)	87.4 ± 5.3	87.7 ± 4.7	87.2 ± 6.0	>0.999
Normal morphology (%)	91.6 ± 6.9	89.1 ± 8.5	93.8 ± 4.2	0.164
DNA damage (Olive Tail Moment)	15.9 ± 3.0	16.6 ± 2.7	15.2 ± 3.2	0.211

**Table 2 animals-12-00204-t002:** Correlation coefficients and the corresponding *p*-values between sperm telomere length and quality parameters.

Parameter	Correlation Coefficient (95% C.I.)	*p*-Value
Progressive motility (%)	−0.018 (−0.429 to 0.395)	0.934
Non-progressive motility (%)	−0.029 (−0.380 to 0.443)	0.896
Fast sperm (%)	−0.326 (−0.812 to 0.335)	0.129
Intermediate sperm (%)	−0.229 (−0.673 to 0.582)	0.292
Slow sperm (%)	−0.311 (−0.663 to 0.078)	0.149
Static sperm (%)	−0.381 (−0.710 to −0.010)	0.073
VCL (µm/s)	0.368 (−0.052 to 0.678)	0.084
VSL (µm/s)	0.076 (−0.347 to 0.473)	0.730
VAP (µm/s)	0.231 (−0.200 to 0.588)	0.288
LIN (VSL/VCL) (%)	−0.015 (−0.424 to 0.400)	0.947
STR (VSL/VAP) (%)	0.226 (−0.205 to 0.584)	0.299
WOB (VAP/VCL) (%)	0.110 (−0.317 to 0.499)	0.618
ALH (µm)	−0.068 (−0.467 to 0.354)	0.757
BCF (Hz)	−0.366 (−0.809 to 0.342)	0.298
Viability (%)	−0.246 (−0.598 to 0.185)	0.257
Normal morphology (%)	0.139 (−0.502 to 0.730)	0.528
DNA damage (Olive Tail Moment)	−0.206 (−0.578 to 0.238)	0.347

**Table 3 animals-12-00204-t003:** In vitro fertilization outcomes for all analyzed samples, and for samples classified into two groups based on their telomere length (shorter or longer than the median); total *n* indicates the total number of embryos including all samples, % represents the average percentage of the parameter analyzed: *p*-values indicate statistical differences of the % between groups with shorter and longer telomere length. (*) means *p* ≤ 0.05.

Parameter	Mean ± SD			*p*-Value
	All Samples	Group with Shorter Telomere Length	Group with Longer Telomere Length	
	Sperm samples = 13	Sperm samples = 6	Sperm samples = 7	
	Oocytes used = 525	Oocytes used = 240	Oocytes used = 285	
Fertilization rate Day 2 (total *n*; %)	177; 34.9% ± 12.3%	72; 30.0% ± 6.9%	105; 39.1% ± 15.5%	0.465
Morulae Day 6 (total *n*; % per sample)	84; 17.0% ± 7.3%	27; 11.5% ± 3.6%	57; 21.8% ± 6.9%	0.018 *
Early blastocysts + blastocysts Day 6 (total *n*; %)	67; 13.1% ± 9.8%	18; 7.6% ± 1.4%	49; 17.9% ± 12.2%	0.018 *
Hatched blastocysts Day 6 (total *n*; %)	10; 1.9% ± 2.6%	1; 0.4% ± 1.0%	9; 3.2% ± 3.1%	>0.999

**Table 4 animals-12-00204-t004:** Correlation coefficients and the corresponding *p*-values between sperm telomere length and in vitro fertilization outcomes. (*) means *p* ≤ 0.05.

Parameter	Correlation Coefficient (95% C.I.)	*p*-Value
Fertilization rate at Day 2 (%)	0.056 (−0.524 to 0.601)	0.856
Morulae at Day 6 (%)	0.559 (−0.007 to 0.854)	0.047 *
Early blastocysts + blastocysts at Day 6 (%)	0.410 (−0.200 to 0.791)	0.164
Hatched blastocysts at Day 6 (%)	0.356 (−0.260 to 0.766)	0.233

## Data Availability

Data generated during the current study are available from the corresponding author on reasonable request.

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
