# Peer review of "Telomere Length in Pig Sperm Is Related to In Vitro Embryo Development Outcomes"

_animals, 2022, doi:10.3390/ani12020204_

Round 1
Reviewer 1 Report
The last few sentences in the abstract are a bit confusing as written, not in regards to results, but in regard to overall conclusions. They seem to downplay the significance of the findings. Suggest more strongly focusing on the significant results (table 3, figure 3) that will have impact in future practical settings vs the correlation results of table 4. The correlation results 'overpower' the other results as it is written now. Also, reiterate these results (table 3) in conclusions (line 460).
a few minor typos/grammatical errors
- in Abstract, use of IVF before it is defined
- Line 75, litter should be plural
- a period in line 338 (should be comma?)
Line 339 - If the authors wish to discuss results based on a tendency towards significance, that tendency should be defined in the statistical section of the methods.
Author Response
Dear Reviewer,
Please find attached our detailed response to your comments in the attached PDF document.
Thank you very much for providing your feedback.

Reviewer 2 Report
The manuscript entitled "The length of telomeres in pig sperm is related to the results of embryonic development" uses a technique commonly used in humans to determine the length of telomeres in porcine sperm. Later they analyze the relationship between telomere length and sperm quality parameters and performance in embryo production. It is a beautiful, novel and important work for the best description of these values in the swine species.
Comments:
- The title should refer to the fact that they are embryos produced in vitro, since many of the alterations in embryos are corrected by the maternal environment and although in vitro they may be significant, they may not be so in vivo.
-Given that the qFISH technique is well developed in humans, it could have been of interest to use a human sperm sample and use it as a control in pigs to verify that all the methodology was adequate.
-Since the result that could be more relevant is to determine how the length of telomeres affects the production of embryos, the in vitro fertilization system should be explained more extensively
- The authors refer that the number of oocytes used per microdroplet during IVF and per male is 40, in the whole experiment have you used only 40 or per replicate? How many replicates?
- An IVF control is not performed to determine polyspermy. It is known that with three pronuclei the embryonic stage is reached.
- It is necessary to determine if the percentage of morulae or blastocysts is based on the number of fertilized or cultured oocytes.
- The total number of oocytes and IVF replications performed is not clear. A minimum of three or four replicates per male used would give more information. This last point is of special relevance since it is the main conclusion of this work.
Author Response

(The authors gave the same response as above.)

Reviewer 3 Report
Comments to the Authors:
The present study approaches the interesting relationship between telomere length, sperm quality and embryo development. However, some questions regarding methodology and statistics have arisen.
Specific comments:
- Lines 68 to 73. Please describe in numbers what is considered short and long telomeres in the studies mentioned.
- Figure 1. The quality of the images shown is low, please increase image quality. Additionally, the sperm head does not seem to be fully decondensed according to the image of the halo provided, and in comparison to the image of the article referenced.
- Please include an image of the particles used for absolute telomere length estimation, as supplementary material.
- “Image captures and analysis”. Image acquisition and processing is essential for reliable fluorescence quantification. Please include a more detailed explanation of how the images were processed, including methods used to correct for background and camera aberrations, deconvolution of the images… Please check the following article as example https://www.nature.com/articles/s42003-019-0692-z.pdf.
- Please indicate how many spermatozoa were analyzed per sample by qFISH.
- Lines 256 to 258. Comparison of number of oocytes used in this study with human studies are irrelevant, since pig ovaries are obtained from slaughterhouse. Please remove the sentence.
- Please clarify if each IVF replicate (using a different boar sample) was performed on a different day with a different pool of oocytes, or several boars were tested on the same day.
- Lines 288 to 289. Please indicate how the percentages of embryo development, morulae, blastocyst…were calculated.
- Line 300. Considering the SD of the telomere length of the different boars, it is difficult to classify them as of long or short telomeres in most cases, since they have spermatozoa falling in both categories. Moreover, taking into account the heterogeneity of the samples, one cannot know if a spermatozoon of “long” or “short” telomeres was the fertilizing one. Therefore, correlation analysis are more relevant than the t tests in this study, since they use the actual telomere length value, rather than a group classification.
- Multiple testing, especially t-tests, are prone to type 1 errors. Please use a correction for multiple testing in all your analysis.
- Line 338. Instead of table 1 it is table 2.
- Table 3. Please indicate the number of oocytes, morulae and blastocyst, included in the study and not only the percentages. Please explain how the percentages are calculated. It seems fairly surprising that the boar samples were chosen randomly and yet you get a 50% of samples with long and short telomeres. Please mention which of the boars were chosen to perform IVF. Why was IVF not performed with all samples?
- Lines 373 to 377. Please treat this statement with caution since there is no statistical significance found for the Spearman test and no correction for multiple comparisons was performed.
- Lines 381 to 383. Since 87% of the samples have values between 18 and 25bp it seems like this is the normal telomere length range in pigs of this age and breed, and the only animals that could be really considered of short and long telomeres are boars 1 and 22-23, respectively.
- Lines 386 to 388. Differences in breed and age should be discussed, besides the different techniques.
- “Relationship between sperm telomere length and quality parameters”. Most of the discussion is based on human results; more weight should be given to other animals, in particular farm animals.
- Lines 441 to 443. I do not know what is the meaning with this sentence, it is also unrelated to the results obtained. Especially considering that, the differences between short and long telomeres are small.
- Lines 443 to 449. The human study mentioned includes a group of normal telomere length. In the present study such a group in not included, only considering long or short telomeres. Therefore, the results of both studies are not comparable. Moreover, there is no significant difference when correlation tests were performed, also no correction for multiple testing was applied. Consequently, this statement should be treated more carefully.
- Lines 460 to 461. This statement is not supported by the presented data and should be removed.
Author Response

(The authors gave the same response as above.)

Round 2
Reviewer 2 Report
I have not more suggestions
Author Response
Reviewers’ comment: I have not more suggestions
Authors’ answer: Thank you very much for your previous feedback, which improved our work.
Reviewer 3 Report
The reviewer would like to thank the authors for the answers provided. However, some of the questions remain unsolved.
Please clarify if each IVF replicate (using a different boar sample) was performed on a different
day with a different pool of oocytes, or several boars were tested on the same day.
Authors’ answer: Thank you for the comment. As for the lab schedule, we conducted experiments in three
batches, including equal samples from each group in each batch. Every day, an excess of ovaries (>30)
was obtained, and a minimum of 250 immature oocytes per sample were harvested. Oocytes were
matured in groups of 50 for two days and, before the IVF procedure, they were pooled for decumulation.
Afterwards, MII oocytes were selected based on morphology under a stereomicroscope, and the best ones
were randomly assigned to one male to perform IVF.
Reviewer´s answer: Taking the answer into account, your samples are not independent, since they are paired by the ovary batch. Therefore, the pairing of the samples must be considered in the statistics.
Table 3. Please indicate the number of oocytes, morulae and blastocyst, included in the study
and not only the percentages. Please explain how the percentages are calculated.
It seems fairly surprising that the boar samples were chosen randomly and yet you get a 50% of
samples with long and short telomeres. Please mention which of the boars were chosen to
perform IVF. Why was IVF not performed with all samples?
Authors’ answer: Thanks for the comment. We included the average number of fertilized embryos,
morulae and blastocysts in Table 3. Regarding sample selection, we randomly chose samples from the
two groups in order to have 50% of samples on each group. We have included an explanation in the
Methods section, as this clarifies our design. We regret that we cannot conduct additional IVF
experiments with the remaining boars, as they are not available anymore from the farm (they were
culled).
Reviewer´s answer: Please include the total number of oocytes, cleaved embryos, morulaes and blastocysts in the table and not the average. It is still not mentioned which of the boars were used for IVF.
Lines 381 to 383. Since 87% of the samples have values between 18 and 25bp it seems like this
is the normal telomere length range in pigs of this age and breed, and the only animals that could
be really considered of short and long telomeres are boars 1 and 22-23, respectively.
Authors’ answer: Thank you for raising this concern. As aforementioned, and after considering the
modification requested in the first comment from this Reviewer, what was considered “long” or “short”
telomere in our study was a simple split of samples considering the median value. If we had taken the
87% of samples as a representative group (samples 2 to 21), the median of values would have been 21.99
kb, which does not much differ from the median obtained when all samples were considered, which was
22.04 kb. If we had used this median, almost all the samples would have fallen within the same
classification to which they now belong to. That being said, we believe that the Reviewer referred to
“outlier samples” rather to samples with longer or shorter telomeres when stating that only animals 1, 22
and 23 should be considered with long or short telomeres.
Reviewer´s answer: What I wanted to point out is the biological significance of the classification in short and long telomeres in this study, since, biologically, there has to be a group of “normal” telomeres, and the difference between an individual with short or long telomeres (as defined in this study) is extremely small. In that context, I am not referring to outliers but to “real” long and short telomeres. That being said, please change in the text long for longer and shot for shorter in all cases. Also, refer to longer or shorter than the median in the abstract and conclusions.
Reviewer´s comment: Line 462- There were no statistical differences in hatching/hatched blastocysts, so please remove it from the sentence.
Author Response
Thank you very much for your feedback. Please find enclosed a .pdf with detailed responses in green colour.
